# Antibiofilm Activity of Phorbaketals from the Marine Sponge *Phorbas* sp. against *Staphylococcus aureus*

**DOI:** 10.3390/md19060301

**Published:** 2021-05-24

**Authors:** Yong-Guy Kim, Jin-Hyung Lee, Sangbum Lee, Young-Kyung Lee, Buyng Su Hwang, Jintae Lee

**Affiliations:** 1School of Chemical Engineering, Yeungnam University, Gyeongsan-si 38541, Gyeongsangbukdo, Korea; yongguy7@ynu.ac.kr (Y.-G.K.); jinhlee@ynu.ac.kr (J.-H.L.); 2Department of Oceanography, Kunsan National University, Kunsan 54150, Korea; sblee08@kunsan.ac.kr (S.L.); leeyk@nnibr.re.kr (Y.-K.L.); 3Nakdonggang National Institute of Biological Resources, 137, Donam 2-gil, Sangju-si 37242, Gyeongsangbukdo, Korea

**Keywords:** antibiofilm activity, marine sponge, phorbaketals, *Staphylococcus aureus*, staphyloxanthin

## Abstract

Biofilm formation by *Staphylococcus aureus* plays a critical role in the persistence of chronic infections due to its tolerance against antimicrobial agents. Here, we investigated the antibiofilm efficacy of six phorbaketals: phorbaketal A (**1**), phorbaketal A acetate (**2**), phorbaketal B (**3**), phorbaketal B acetate (**4**), phorbaketal C (**5**), and phorbaketal C acetate (**6**), isolated from the Korean marine sponge *Phorbas* sp. Of these six compounds, **3** and **5** were found to be effective inhibitors of biofilm formation by two *S. aureus* strains, which included a methicillin-resistant *S. aureus*. In addition, **3** also inhibited the production of staphyloxanthin, which protects microbes from reactive oxygen species generated by neutrophils and macrophages. Transcriptional analyses showed that **3** and **5** inhibited the expression of the biofilm-related hemolysin gene *hla* and the nuclease gene *nuc1*.

## 1. Introduction

Most microbes can form biofilms on living or abiotic surfaces, and such biofilms are common in environmental and medical settings [1]. Biofilms encased by extracellular polymeric substances reduce microbial sensitivity to antimicrobial agents, host defenses, and external stresses, and thus, contribute to bacterial persistence and chronic infections [2]. Furthermore, subinhibitory concentrations of several antibiotics often increase biofilm formation, because they trigger bacterial defense systems [3], which means bacterial biofilm formation is a mechanism of antibiotic resistance. Thus, it is important that novel compounds capable of inhibiting biofilm formation without bactericidal effects be identified to mitigate the challenges posed by drug resistance.

Biofilm formation by *Staphylococcus aureus* is of particular concern in the medical field, and methicillin-resistant *S. aureus* (MRSA) and vancomycin-methicillin-resistant *S. aureus* are serious agents of nosocomial infections [4]. Furthermore, *S. aureus* can secrete exotoxins, such as hemolysin, enterotoxins, coagulase, and TSST-1, which are associated with its pathogenesis, and can evade host immune systems in various ways, such as by producing the carotenoid staphyloxanthin, which helps microbes survive in the presence of reactive oxygen species generated by neutrophils and macrophages [5].

Marine sponges are rich sources of bioactive compounds, notably antibiotics, enzymes, enzyme inhibitors, and pharmacologically active agents [6]. Many marine sponges contain dense microbial communities, including diverse bacteria and archaea [7]. It appears that sponges and microbes coexist due to balanced host–microorganism interactions, and marine sponges have been shown to produce various small molecules that modulate bacterial biofilm formation [8]. For example, there are two classes of marine sponge metabolites, namely terpenoids and pyrrole-imidazoles, that are considered non-bactericidal biofilm inhibitors. Additionally, it has been reported that phorbaketals A, B, and C, isolated from the marine sponge *Phorbas* sp., exhibit moderate cytotoxicity against human colorectal, hepatoma, and lung cancer cell lines [9]. Furthermore, phorbaketal A has been reported to have anti-inflammatory effects [10] and to stimulate osteoblast differentiation [11]. However, no information on the antibiofilm activity of phorbaketals has been reported.

In this study, six phorbaketals were isolated from the Korean marine sponge *Phorbas* sp. and evaluated for their antibiofilm activities against two *S. aureus* strains, including a MRSA strain. We also investigated the effects of phorbaketals on staphyloxanthin production and studied the antibiofilm mechanisms of the most active compounds using a transcriptomic approach.

## 2. Results and Discussion

Six phorbaketals, namely, phorbaketal A (**1**), phorbaketal A acetate (**2**), phorbaketal B (**3**), phorbaketal B acetate (**4**), phorbaketal C (**5**), and phorbaketal C acetate (**6**) (Figure 1), were successfully isolated from the Korean marine sponge *Phorbas* sp. Previously reported compounds **1**, **3**, and **5** were identified as phorbaketal A, B, and C, respectively, by comparing their spectroscopic profiles (NMR spectra and HR-ESIMS profiles are shown in Appendix A) with published data [10].

Compound **2** was isolated as a yellowish oil, and its molecular formula was determined to be C_27_H_36_O_5_ by HRESIMS ([M + Na]^+^ ion peak *m*/*z* 463.2460), which was consistent with ten degrees of unsaturation. The IR spectrum of **2** showed a strong absorption band at 1743 and 1226 cm^−1^, indicating the acetyl group and its UV spectrum had an absorption band at 203 nm (ɛ 26480). This compound had a specific optical rotation value ${\left[\alpha \right]}_{D}^{25}$ of −63.9 (c = 0.15, MeOH). The ^1^H and ^13^C NMR spectra of **2** demonstrated the presence of six methyl, five methylene, and eight methine groups, and eight quaternary carbons. In particular, five singlet resonances in the range 1.61 to 1.81 ppm indicated the presence of olefinic methyls, and one methyl peak at 2.06 ppm indicated carbonyl carbon at 172.3 ppm. Furthermore, ten chemical shifts of sp^2^ carbons indicated the presence of five double bonds, which suggested that **2** possessed three rings.

The COSY, TOCSY, and HSQC spectra of **2** were similar to that of phorbaketal A (**1**) (Figure 2). Connectivity of the parts of **2** (a–c) was secured by analyzing HMBC spectroscopic data (Appendix A). The presence of an acetyl group was evident from the presence of a methyl proton peak at δ 2.06 ppm and HMBC cross-peaks H-2′/C-1′ and C-9 (Figure 2A). The stereochemistry of **2** was established by ROESY and using coupling constants. NOE correlations (H-1/H-7, and H-19; H-6a/H-12; H-10/H-12) of **2,** as displayed in Figure 2B, indicated the same relative configuration as that reported for **1**. Furthermore, the coupling pattern of H-6a (δ_H_ 2.44, dd, *J* = 14.2, 11.7 Hz), which showed large coupling constant values, determined the configuration of H-6. Thus, the structure of phorbaketal A acetate (**2**) was determined, as shown in Figure 1.

The molecular formula of **4** was determined to be C_25_H_38_O_5_ based on an HRESIMS molecular ion peak at m/z 465.2611 [M + Na]^+^ and ^13^C NMR data. The ^1^H and ^13^C NMR spectra of **2** resembled that of phorbaketal B (**3**), a previously reported natural product, except that **3** contained a methyl proton and one additional carbonyl carbon. HMBC correlations showed the carbonyl carbon (δ_C_ 172.4) was attached to a methyl proton (δ_H_ 2.06) and an oxy-methine proton H-5 (δ_H_ 5.37), indicating **4** had an acetyl group at C-5 (Figure 1). The stereochemistry of **4** was assigned using *J* values in its ^1^H NMR spectrum and NOESY data. The coupling constants of H-6a (ddd, *J* = 13.2, 11.7 and 10.7 Hz) observed in the ^1^H NMR spectrum were similar to those of H-6a (ddd, *J* = 12.5, 11.5 and 10.3 Hz) in **3**, with three large coupling constant values, indicating H-6a was configured in the *anti* position with two vicinal protons; H-5 (δ_H_ 5.37) and H-7 (δ_H_ 2.17). The stereochemistry of H-5 was assigned to the *R* form, and this was supported by definite NOE correlations between H-7 and H-5. Therefore, compound **4** was determined to be phorbaketal B acetate.

Phorbaketal C acetate (**6**) possessed the same molecular formula as **4** (C_25_H_38_O_5_) based on a molecular ion peak [M + Na]^+^ at *m*/*z* 465.2614 (calcd for C_25_H_38_O_4_Na, 465.2611) in its HRESIMS spectrum. The only differences between **4** and **6** were chemical shifts at C-5 (δ_H_ a 1.55, b 2.11/δ_C_ 73.2 in **4**; δ_H_ a 1.72, b 1.95/δ_C_ 71.1 in **6**), which suggested **4** and **6** were stereoisomers at C-5. This was confirmed by a coupling constant value of H-6b (ddd, J = 14.4, 4.0, and 3.1 ppm), which showed **6** was configured differently at this position. Hence, three acetylated phorbaketals, **2**, **4**, and **6**, are successfully purified in this study.

The six phorbaketals were initially screened for the inhibition of methicillin-sensitive *S. aureus* (MSSA) ATCC 6538 biofilm formation in 96-well microtiter plates at concentrations of 0, 10, 20, 50, or 100 μg/mL. All six phorbaketals significantly inhibited biofilm formation in a dose-dependent manner, though **3** and **5** had the greatest inhibitory effects (Figure 3A). Similar antibiofilm activities were observed against MRSA MW2 (Figure 3B), though the MSSA strain was more susceptible.

Ideally, an antibiofilm agent should not kill bacteria, because this would probably lead to bacterial drug resistance. Thus, the antimicrobial activities of the six phorbaketals were assessed by measuring minimum inhibitory concentrations (MICs). The MICs of all six compounds against *S. aureus* 6538 strain exceeded 200 μg/mL, indicating mild antimicrobial activities. Additionally, the planktonic cell growth of *S. aureus* was relatively unaffected by the six phorbaketals at <100 μg/mL (Appendix A). These results show that the six phorbaketals at sub MIC concentrations reduced biofilm formation because of their antibiofilm activities and not their antimicrobial activities. Because compounds **3** and **5** reduced biofilm formation most and did not affect planktonic cell growth, we focused on these two compounds in our mechanistic study.

Confocal laser scanning microscopy was also used to observe changes in biofilm formation. In line with our quantitative biofilm data (Figure 3), fluorescence images indicated that **3** and **5** at 100 µg/mL markedly inhibited biofilm formation by *S. aureus* MSSA 6538 strain (Figure 4A). Changes in biofilm formation were also quantified using COMSTAT biofilm software. The results obtained show that **3** and **5** reduced the biomasses (volume/area), mean thicknesses, and substratum coverages of *S. aureus* biofilms by >95% (Figure 4B).

In addition, we investigated the effects of the six phorbaketals on the production of staphyloxanthin (a yellow pigment). Staphyloxanthin production was identified visually in cell pellets of untreated *S. aureus*, and compounds **3**, **4**, and **5** reduced staphyloxanthin production after culture for 6 h, but only compound **3** continued to reduce staphyloxanthin production after culture for 24 h (Figure 5), which suggested that **3** would diminish the ability of *S. aureus* to evade host immune systems. Visual observation can be further quantified with a quantitative assay [12].

To investigate the molecular mechanisms responsible for the inhibitory effects of phorbaketals on biofilm formation by *S. aureus*, we investigated the expression of selected biofilm- and virulence-related genes and important regulatory genes by qRT-PCR. Notably, **3** and **5** dramatically reduced the expression of α-hemolysin (*hla*) and nuclease (*nuc1*), though they had smaller effects on the expression of other biofilm-related genes (Figure 6). Additionally, **5** significantly reduced the expression of RNAIII (a regulatory RNA molecule). It has been reported that RNAIII stimulates *hla* translation [13]. Hence, *hla* repression was attributed to the downregulation of *RNAIII* by **5**. Furthermore, *hla* has been reported to play an important role during biofilm formation [14], and the downregulation of *hla* in *S. aureus* has been observed in the presence of other antibiofilm agents, such as azithromycin [15], stilbenoids [16], flavonoids [17], norlichexanthone [18], or alizarin [19]. Additionally, omega fatty acids were recently reported to repress the expression of *hla* and *RNAIII* and inhibit biofilm formation [20]. In line with previous reports, compounds **3** and **5** inhibited *S. aureus* biofilm formation, at least in part, by downregulating the expression of the α-hemolysin *hla* gene and *RNAIII*. The involvement of α-hemolysin in biofilm inhibition could be further confirmed by the assay with exogenous addition of α-hemolysin [21]. The downregulation of *nuc1* by **3** and **5** was unexpected, since nucleases prevent *S. aureus* biofilm formation [22,23]. On the other hand, mutations of nuclease genes (*nuc1* and *nuc2*) were reported to enhance biofilm formation [24]. Based on currently available transcriptomic data, the roles played by *S. aureus* nucleases during biofilm formation under *in vitro* conditions remain unclear.

Several natural compounds containing hydroxyl bipyridine, hydroxyl anthraquinone, or hydroxyl flavonoid structures, such as collismycin C [25], alizarin, purpurin, emodin [19], fisetin, luteolin, and quercetin [17], have been reported to exhibit antibiofilm activity against *S. aureus*. Interestingly, the antibiofilm activities of collismycin, anthraquinones, and flavonoids were found to be related to the number and positions of hydroxyl groups in backbone structures. In this study, among the six phorbaketals studied, compounds **3** and **5** (an isomer of **3**), which possessed two hydroxyl groups, more potently inhibited *S. aureus* biofilm formation than the other phorbaketals (Figure 3). Thus, it appears that the presence and positions of hydroxyl groups on phorbaketals importantly determine antibiofilm activity against *S. aureus*.

This study demonstrates for the first time that phorbaketals **3** and **5** inhibit *S. aureus* biofilm formation and staphyloxanthin production by *S. aureus*, and do so by downregulating the expression of the hemolysin-related genes *hla* and *nuc1*. Compounds **3** and **5** have been previously reported to be only moderately cytotoxic to human epithelial HT-29 cells (IC_50_ 28 and 212 µg/mL) [9]. Thus, it appears phorbaketals **3** or **5** are potential lead compounds for antibiofilm agents against drug-resistant *S. aureus*.

## 3. Materials and Methods

### 3.1. General Experimental Procedures

A specimen of *Phorbas* sp. (voucher number 07G-26) was collected by hand from Gageo Island, South Korea, in 2007. Optical rotations were measured on a JASCO P-1010 polarimeter (Jasco, Easton, MD, USA). IR spectra were recorded using a JASCO FT/IR 4100 spectrometer (Jasco, Easton, MD, USA), and ultraviolet (UV) spectra using a Varian Cary 50 UV-Visible spectrophotometer (Agilent, Santa Clara, CA, USA). High-resolution (HR)-electrospray ionization (ESI) mass spectra were obtained using a SCIEX X500R mass spectrometer (Sciex, Framingham, MA, USA). The NMR spectra were recorded on a Varian VNMRS 500 NMR spectrometer (Varian, Palo Alto, CA, USA) operating at 500 (^1^H) or 125 MHz (^13^C), respectively, with chemical shifts given in ppm using a methanol-*d*4 solution concerning residual solvent peaks at 3.30 and 49.0 ppm. Semi-preparative liquid chromatography was performed using an Agilent 1200 pump (Agilent, Santa Clara, CA, USA) and an RI detector. Vacuum column chromatography was performed using RP-18 silica gel 60 (Merck, Darmstadt, Germany).

### 3.2. Isolation of Phorbaketals from Phorbas sp.

The collected specimen was frozen on site, transported to a laboratory under dry ice, and kept in a refrigerator at −25 °C until required. The lyophilized sponge was extracted twice at room temperature with MeOH and filtered. The methanolic extract was partitioned between dichloromethane (DCM) and distilled water, and then the organic layer was repartitioned between *n*-hexane and 15% aqueous MeOH for defatting. The aqueous MeOH fraction (ca 4.5 g) was performed on the vacuum column chromatography eluted with seven different solvent mixtures of decreasing polarity (MeOH/H_2_O = 5/5; 6/4; 7/3; 8/2; 9/1; 100% MeOH; 100% acetone) to give seven fractions (MR1–MR7). Reversed-phase HPLC using a MeCN/H_2_O (65/35) solvent and a C18 semi-prep column (YMC ODS-A column, 250 × 10 mm ID) at a flow rate of 2 mL/min separated fraction MR5 (1.4 g), yielding compounds **3** (50 mg), **4** (68 mg), **5** (15.5 mg), and **6** (69 mg). Compounds **1** (320 mg) and **2** (166 mg) were isolated from fraction MR6 (970 mg).

### 3.3. Strains and Culture Conditions

*S. aureus* MSSA ATCC 6538 and MRSA ATCC BAA-1707 MW2 were used in the study. All experiments were conducted at 37 °C. Bacteria were initially streaked from −80 °C glycerol stocks onto LB plates, and a single fresh colony was inoculated into 25 mL LB medium in a 250 mL flask and cultured overnight at 37 °C with shaking at 250 rpm. Overnight cultures were re-inoculated into LB medium at a dilution of 1:100. Cell growths in the presence of different concentrations of compounds were monitored by measuring absorbance at 620 nm (OD_620_) using a spectrophotometer (UV-160, Shimadzu, Japan). Minimum inhibitory concentrations (MICs) were determined (Appendix A) as specified by the Clinical Laboratory Standards Institute (CLSI) for bacteria and yeasts [26]. All experiments were performed using at least two independent cultures.

### 3.4. Antibiofilm Assays

Biofilm inhibitory effect was measured using crystal violet as previously described [20]. Briefly, *S. aureus* cells were inoculated into LB medium (for the MSSA strain) or LB supplemented with 0.2% glucose (for the MRSA strain) at an initial OD_600_ of 0.05 in a total volume of 300 µL in 96-well polystyrene plates (SPL Life Sciences, Pocheon, Korea). Cells were then cultured with or without phorbaketals (0, 10, 20, 50, and 100 µg/mL) for 24 h without agitation. Biofilm cells were stained with 0.1% crystal violet for 20 min, washed three times with distilled water, and dissolved in 95% ethanol, and absorbance at 570 nm (OD_570_) was measured using a Multiskan EX microplate reader (Thermo Fisher Scientific, Waltham, MA, USA). Cell growth in 96-well plates was also monitored by measuring absorbance at 620 nm (OD_620_). Results represent the means of at least 12 replicate wells.

### 3.5. Confocal Laser Microscopy and COMSTAT Analysis

Biofilm formation was also visualized by confocal laser microscopy (Nikon Eclipse Ti, Nikon Instruments, Tokyo, Japan) using an Ar laser (excitation 488 nm, emission 500–550 nm) and a 20× objective. Color confocal images were produced using NIS-Elements C version 3.2 (Nikon Instruments). For each experiment, at least 10 random positions in each of the three independent cultures were chosen for microscopic analysis. To quantify biofilm formation in the presence and absence of phorbaketals (100 µg/mL), COMSTAT biofilm software (kindly provided by Arne Heydorn, Søborg, Denmark) was used to determine biomasses (μm^3^/μm^2^), mean thicknesses (μm), and substratum coverage (%).

### 3.6. Staphyloxanthin Assay

The bright golden color of staphyloxanthin enabled simple pellet observation [27]. Briefly, *S. aureus* MSSA 6538 cells were inoculated at 1:100 dilution in LB (2 mL) and incubated for 16 h with phorbaketals (100 µg/mL), at 37 °C in 14-mL tubes at 250 rpm. Cells (1 mL) were then collected by centrifugation at 16,600× *g* for 10 min and cell pellets were photographed to show staphyloxanthin production.

### 3.7. Quantitative Real-Time Reverse Transcription (qRT-PCR) Analysis

RNA isolation and qRT-PCR was performed, as previously described with slight modification [19]. Briefly, to isolate total RNA, *S. aureus* MSSA 6538 cells were inoculated into 25 mL of LB broth in 250 mL flasks at an OD_600_ of 0.05 and then incubated for 6 h at 37℃ with shaking at 250 rpm in the presence or absence of phorbaketal B (**3**) or phorbaketal C (**5**) at 100 μg/mL. RNase inhibitor (RNAlater, Ambion, TX, USA) was then added, and cells were immediately chilled for 30 s in a dry ice bath containing 95% ethanol to prevent RNA degradation. Cells were then harvested by centrifugation at 16,600× *g* for 1 min, and total RNA was isolated using a Qiagen RNeasy mini Kit (Valencia, CA, USA). qRT-PCR was used to determine the transcriptional levels of 10 biofilm-related genes, viz *agrA*, *arlS*, *aur*, *hla*, *icaA*, *nuc1*, *RNAIII*, *saeR*, *sarA*, *sarZ*, *seb*, *sigB*, and *spa*. Gene-specific primers were used and 16s rRNA was used as the housekeeping control (Appendix A). qRT-PCR was performed using a SYBR Green master mix (Applied Biosystems, Foster City, CA, USA) and an ABI StepOne Real-Time PCR System (Applied Biosystems). Expression levels were determined using two independent cultures and six qRT-PCR reactions were performed per gene.

### 3.8. Statistical Analysis

At least two independent cultures were used and replication numbers are presented above. The statistical analysis was performed using one-way ANOVA followed by Dunnett’s test using SPSS version 23 (SPSS Inc., Chicago, IL, USA). *p* values of <0.05 were considered significant, which is indicated using asterisks.

## Figures and Tables

**Figure 1 marinedrugs-19-00301-f001:**
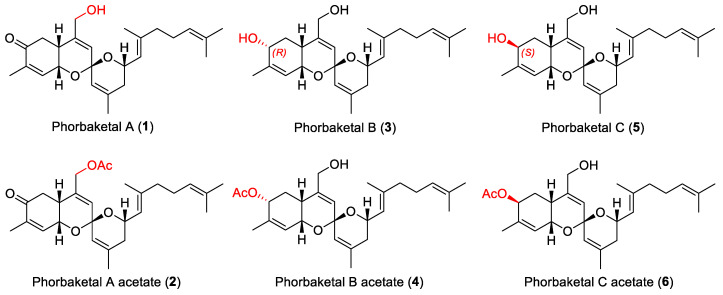
The structures of compounds **1**–**6**.

**Figure 2 marinedrugs-19-00301-f002:**
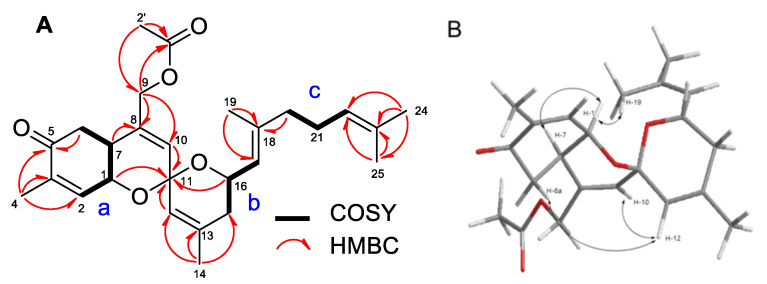
The COSY and key HMBC correlations (**A**), NOE correlations of compound **2** (**B**).

**Figure 3 marinedrugs-19-00301-f003:**
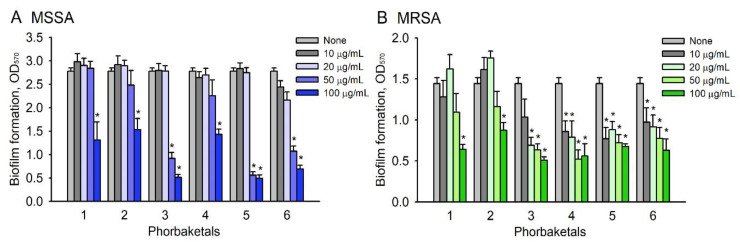
Inhibition of biofilm formation by phorbaketals. Biofilm formations (OD_570_) by *S. aureus* MSSA 6538 (**A**) and *S. aureus* MRSA MW2 (**B**) were evaluated after incubation for 24 h in 96-well plates in the presence or absence of phorbaketals (**A**). *, *p* < 0.05 versus untreated controls.

**Figure 4 marinedrugs-19-00301-f004:**
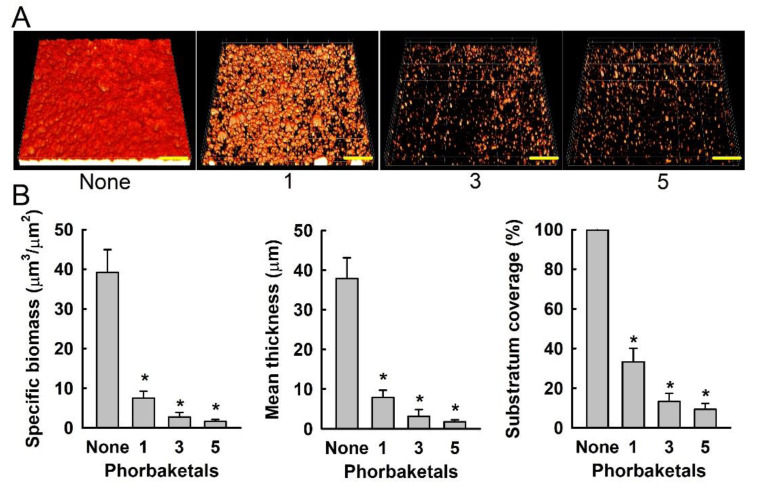
(**A**) Confocal laser scanning microscopy images of biofilm formation in the presence of phorbaketals **1**, **3**, or **5** at 100 µg/mL. Scale bars represent 100 μm. (**B**) COMSTAT quantifications of biofilm formation. * *p* < 0.05 versus untreated controls.

**Figure 5 marinedrugs-19-00301-f005:**
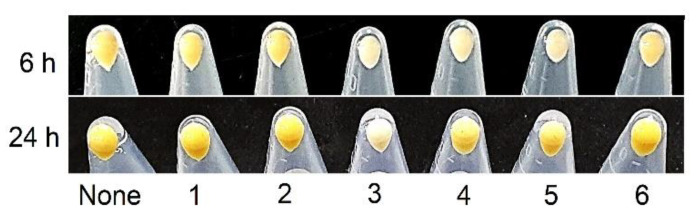
Effects of phorbaketals at 100 µg/mL on staphyloxanthin production in *S. aureus*.

**Figure 6 marinedrugs-19-00301-f006:**
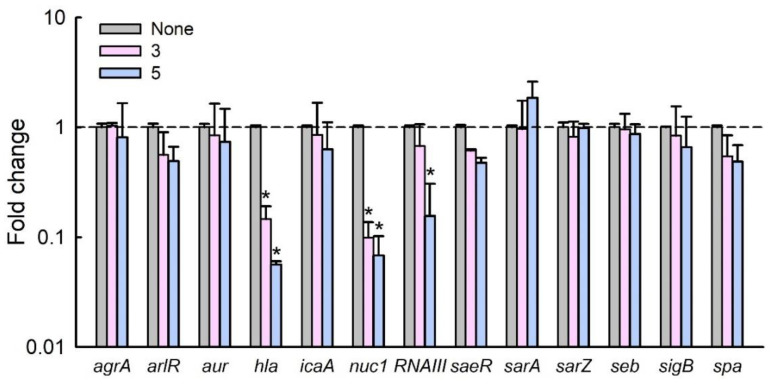
Transcriptional profile changes in *S. aureus* MSSA ATCC 6538 after treatment with compounds **3** or **5.** Relative transcriptional profiles were determined by qRT-PCR with respect to 16s rRNA expression. *, *p* < 0.05 versus untreated controls.

## Data Availability

Data are contained within the article.

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
