# Peer review of "Antibiofilm Activity of Phorbaketals from the Marine Sponge *Phorbas* sp. against *Staphylococcus aureus"

_marinedrugs, 2021, doi:10.3390/md19060301_

Round 1

Reviewer 1 Report

This article describes the antibiofilm activity of six phorbaketals against Staphylococcus aureus. The topic is of high relevance and phorbaketals seem to be a promising new class of antibiofilm agents. However, there remain a few open questions in regard to experimental details of this manuscript.

Major

  • The authors should clearly state which statistical analysis they performed.
  • Line 123ff: The authors state that the MICs were >200 µg/mL. However I cannot find data for 200µg/mL presented in Fig S.18.
  • Lines128ff: The authors claim that substance 3 and 5 reduced biofilm growth the most and only continued with these substances. However, based on the data presented in Fig. 3 I do not see any evidence that substance 6 (and maybe 4) is less effective. I also want to point out that the authors indeed performed subsequent experiments with substance 1 (Fig. 4) and all six substances (Fig. 5). The authors should clearly state their reasoning for this selective presentation of data. Especially as all of the substances inhibited biofilm growth to some degree. I suggest that the authors should also perform gene expression experiments for substance 1, in order to have at least one representative of each phorbaketal included in all experiments.
  • Lines 146: Given the significance of the statements, I would suggest that the authors use a more quantitative assay to give their findings more impact. There are easy spectroscopic assays for the quantification published (e.g. https://doi.org/10.1038/s41598-020-79976-7).
  • Lines 182: I agree with the authors that the downregulation of hla should influence biofilm formation. However, the authors did not prove this connection in this study. The authors could perform biofilm rescue experiments with the addition of for exogenous AT (e.g. 3390/toxins10040157) in order to clarify this point.

Minor:

  • Figure 4. The numbering of the phorbaketals is inconsistent in the caption and the figure.
  • Line 125ff: Based on the data in the Fig S18, only substances 3 and 5 showed any negative impact on the growth curve. However, it is hard to judge how significant the results are, given the fact that experimental details are missing.

Author Response

Our response is attached.

Reviewer 2 Report

The research carried out by the authors seemed to be interesting, however, this aspect of antibiofilm activity of marine organism-derived products has been studied for many years.  Therefore, the authors should particularly emphasize the new approach to the problem and underline the obtained original information that goes beyond the earlier available reports. One could expect more deep knowledge than studying the impact of known products (which chemical structure is mostly known and have been published by one of the co-author in 2009) on biofilm formation by only two bacterial strains, using simple lab method without assessing and describing the proven mechanisms of the observed phenomenon.

General comments The current MS version is not devoid of numerous limitations. The obtained results do not entitle to present far-reaching conclusions of the authors.

Some critical remarks are given below. The authors should consider the following specific comments.

Introduction section presents and justifies the purpose of research in very general terms. There is no so-called "fluidity" of the text, so the content should be re-written. For example info from L47-50 is not enough convincing to repeat known experiments. On the other hand, text from L 50-54 contains data not related to the purpose of the research, thus it should be extended if it is in line with purpose of the study or deleted.

In general, the so-called guiding idea of the study is missing, which should be an essential part of any original study report.

Material and Methods

The current description of the research methods used does not allow them to be repeated in another laboratory, which is an essential condition for their reliability. They should necessarily be described in a more accurate version.

-Subsection 3.1, 3.2 and Supplementary Material contain mostly data already published by one of the coo-author in 2009.

-Subsection 3.3, 3.4,3.5 is described identically as in published own works, even duplicating errors.-Why it was decided to use collismycin C (L247) in research?

- Why were only 2 strains of S.a. since it is known that the phenotypic diversity of staphylococci is very large. This applies to the synthesis of enzymes, toxins, surface proteins as well as the formation of a biofilm, susceptibility to various external factors, etc.

-Subsection 3.6- Why staphylxanthin production has been studied - this is just one of many products involved in the interference of staphylococci with defense mechanisms. Moreover, the subjective qualitative method of staphylxanthine production evaluation is not sufficient to assess the effect of the tested products.

- Subsection 3.7- Why has MRSA not been tested

The Results and Discussion section lacks an in-depth discussion of the meaning of the less original observations made by the authors. The list of references does not include many publications from recent years, closely related to the subject of the conducted research.

- Were products 1-6 isolated from sponges devoid of inhabiting microorganisms? It is reported that they were harvested in 2007 and kept frozen. So, what is the difference between the now written isolation and characterization procedure and that published by Rho et al. In 2009?

- The results on biofilm formation shown in Fig. 3 obtained by the CV biomass staining method are not sufficiently reliable. Has inhibition of bacterial growth, decreased EPS production, or some other effect been achieved? Please provide the OD620 results reported on line 238-240.

- The description of Fig. 4. A and B is not consistent with the presented graphic documentation (products 1,3,5 or 1,2,3 were tested?).

- The conclusion that its reduced level is of great importance for the pathogenicity of S.a. (L 143-147) goes too far.

- L150-171 - The results of the molecular analysis concern 1 strain of S.a. - which one and why this.

- Based on the sparse experimental data presented, the conclusions in L182-188 are not valid.

- The list of references does not include many publications from recent years, closely related to the subject of the conducted research. Self-citations are too numerous and unjustified

Author Response

Our response is attached.

Reviewer 3 Report

Dear Authors,

This study described the exploration of a sponge Phorbas sp. collected in South Korea, which led to the isolation of three known metabolites, phorbaketals A-C along with three acetylated phorbaketals A-C. In addition, compounds 3 and 5 exhibited promising antibacterial activities against MRSA.

1) Page 2, line 65, "compounds 1, 3, and 5 spectroscopic profiles (NMR, UV, IR, MS) (Figures S1-S17)". However, Fig S1-S17 is NMR spectra for 2, 4 and 6, therefore it will misled reader that Fig S1-S17 is for 1, 3 and 5.

2) Page 2, line 72, "IR 1226 cm-1 indicating a carbonyl group", I think IR 1226 cm-1 suggested the presence of alkoxy functionality.

3) Page 6, section 3.2, lines 214-219, the yield of isolated compounds 1-6 should be reported.

4) Page 6, section 3.2, line 219, did Authors carried out Mosher's method to determine the absolute configuration of isolated compounds in this work? If not, this sentence has to be removed to avoid confusion.

5) Page 6, section 3.2, line 218, compounds 1, 3, and 5 were identified as known metabolites by comparison of spectroscopic data, which including NMR, UV, IR and MS. But in supplementary material, UV and IR result were not included. If it was not measured, Authors have omit the comparison of UV and IR in the manuscript. 

6) Supplementary material, page 21, Table S1, compound 4: H-15a 1.83 dd (17.3, 3.3); H-15b 1.99 br d (11.1). I think something is missing, H-15b 1.99 br d is 11.1 or 17.3 Hz? because the germinal coupling constant has missing.

7) Supplementary material, page 21, Table S1, compound 6: splitting pattern of H2-15 and H-17 was described as dm, I think this is uncommon, Authors might consider revise this.

8) Supplementary material, page 21, Table S1, compound 6: H2-9 has a individual signals assigned at 4.02 and 4.05 ppm. While, isolated compound 5 and reported phorbaketal C has a superimposed signal at 4.08 ppm. All these compounds have identical stereochemistry, I think H2-9 in 6 should behave similar to those of 5 and reported phorbaketal C.

9) Page 3, Figure 2B, the rings A, B and C wasn't really a chair conformation due to the presence of double bond as part of the 6-membered ring. Please see the below reference to redraw the figure 2B.

Phorbaketals A, B, and C, sesterterpenoids with a spiroketal of hydrobenzopyran moiety isolated from the marine sponge Phorbas sp. Org. Lett. 2009, 11, 5590-5593

10) Page 5, Figure 5, at 24 h, staphyloxanthin production looked drastically dropped only with the treatment of 3 but not others, it can be understandable that further investigation was carried out on 3 together with 5 as 5 is the 5-OAc of 3. However, Fig 5 showed staphyloxanthin production in 4 and 5 looked almost same comparing to more yellowish 1, 2 and 6. Why 4 was not selected for further investigation?

11) Presence of OAc unit is kind of skeptical as being a natural products, as many natural products possessed OAc moieties were in fact an artifact. What are the chances to exclude this possibility as an artifacts for 2, 4 and 6?

Author Response

Our response is attached.

Round 2

Reviewer 1 Report

The changes made by the authors improved the manuscript, however it is still my opinion that additional experiments should be performed and my response to the author’s responses are listed below. I am sure the editorial staff will agree with an extended timeline to perform these experiments.

“Lines 146: Given the significance of the statements, I would suggest that the authors use a more

quantitative assay to give their findings more impact. There are easy spectroscopic assays for the

quantification published (e.g. https://doi.org/10.1038/s41598-020-79976-7).

Thank you for introducing this interesting reference of quantification of staphyloxanthin

production. We have cited this paper (Line 147) and it is shown as a similar matter in images (Fig.

5)“

- My original point remains, that a spectroscopic analysis would allow the authors a more detailed analysis of possible inhibition of staphyloxanthin production by compound 3. Analysis as performed by (https://doi.org/10.1016/j.fct.2018.06.017) would allow also for the detection and quantification of intermediates of staphyloxanthin production. Given the structural similarity of the studied compounds and the very qualitative analysis, it remains unclear what differentiates the different compounds.

“We appreciate the Reviewer’s insightful comments. We agree that all six phorbaketals inhibited S.

aureus biofilm formation to some degree in a dose-dependent manner (Fig. 3A and 3B). Initially,

we tested all six substances against two S. aureus strains (Fig. 3) and selected the two best

phorbaketals 3 and 5 based on stronger inhibitory activity compared to other phorbaketals at 100

μg/mL (Fig. 3A). Also, all six phorbaketals were investigated for staphyloxanthin production (Fig.

3 5) and the antibiofilm activity of three phorbaketals, 1, 3, and 5 were confirmed by confocal laser

scanning microscopy. Then, we wanted to focus on the most active hits 3 and 5 and further studied

the transcriptomic study (Fig. 6). It is expected that phorbaketal 1 would show less change of gene

expression than 3 and 5.”

- I do understand the reasoning of the authors for focusing on only a few compounds. However, the fact that compound 1 was included in the confocal scanning microscopy experiments, but not in the gene expression analysis remains unsatisfactory.  As currently presented, the results and conclusions have a lot of “maybe´s” attached to them. It is my opinion that the potential agreement of the reduced biofilm inhibition with less reduced gene expression (e.g. of hla) would strengthen the authors conclusions. Ideally this would be performed with all six phorbaketals in order to determine the effect of the acetate, but should be at least be done with compound one.

Author Response

We appreciate the Reviewer’s insightful comments. Response was attached below.

Reviewer 2 Report

I accept the amendments. They cover most (but not all) of my comments and concerns. I have no further serious comments.

Author Response

We appreciate the reviewer’s understanding and endorsement.

Reviewer 3 Report

Dear Authors,

Thank you for addressing the comments positively. However, there are some typing mistakes.

Page 2, line 68, HRFABMS should be HRESIMS. 

Page 4, line 139, phorbaketals 1, 3, or 5 should be phorbaketals 1, 3, or 5

Author Response

We appreciate the reviewer’s understanding and other corrections. As suggested, two typing mistakes have been corrected.

Round 3

Reviewer 1 Report

As the authors pointed out in their response, we are in disagreement over some of the experiments.

I want to point out that I do believe that the suggested experiments would strengthen the presented results and their impact but might not reveal any new insights. Therfore I do understand the authors reasoning but I do find it unfortunate.

Given the fact that the authors improved the manuscript according to all other comments, I have nothing to add.